# Spatial Distribution of Equid Exposure to *Rickettsia* spp. in Goiás State, Midwestern Brazil

**DOI:** 10.3390/pathogens14050449

**Published:** 2025-05-02

**Authors:** Gracielle Teles Pádua, Mariana Avelar Tavares, Nicolas Jalowitzki de Lima, Warley Vieira de Freitas Paula, Gabriel Cândido dos Santos, Lucianne Cardoso Neves, Raphaela Bueno Mendes Bittencourt, Raquel Loren dos Reis Paludo, Ennya Rafaella Neves Cardoso, Bianca Barbara Fonseca da Silva, Bruno Rodrigues de Pádua, Ana Carolina Borsanelli, Filipe Dantas-Torres, Gina Paola Polo, Felipe da Silva Krawczak

**Affiliations:** 1Laboratório de Doenças Parasitárias—LADOPAR, Setor de Medicina Veterinária Preventiva, Escola de Veterinária e Zootecnica, Universidade Federal de Goiás—UFG, Goiânia 74690-900, Brazil; gracielletelespadua@egresso.ufg.br (G.T.P.); mariana.tavares@discente.ufg.br (M.A.T.); jalowitzki@discente.ufg.br (N.J.d.L.); warleyvieira@egresso.ufg.br (W.V.d.F.P.); doscandido@discente.ufg.br (G.C.d.S.); luciannecardoso@egresso.ufg.br (L.C.N.); rafabmbitt@discente.ufg.br (R.B.M.B.); raquelloren@unifimes.edu.br (R.L.d.R.P.); ennyaneves@discente.ufg.br (E.R.N.C.); biancabarbarafs@gmail.com (B.B.F.d.S.); 2Agência Goiana de Defesa Agropecuária, Goiânia 74620-030, Brazil; bruno.rpadua@goias.gov.br; 3Laboratório de Doenças Infecciosas e Bacteriologia—LADIB, Setor de Medicina Veterinária Preventiva, Escola de Veterinária e Zootecnica, Universidade Federal de Goiás—UFG, Goiânia 74690-900, Brazil; anaborsanelli@ufg.br; 4Departamento de Imunologia, Centro de Pesquisa Aggeu Magalhães, Fundação Oswaldo Cruz, Recife 50670-420, Brazil; filipe.torres@fiocruz.br; 5Instituto de Salud Pública, Pontificia Universidad Javeriana, Bogotá 110231, Colombia; imaginapolo@gmail.com

**Keywords:** *Amblyomma sculptum*, *Dermacentor nitens*, horses, *Rickettsia amblyommatis*

## Abstract

This study sought to investigate the presence of anti-*Rickettsia* spp. antibodies in georeferenced serum samples from equids across all regions of the state of Goiás, while also presenting variables that indicate risk factors for the circulation of rickettsiae, and evaluating the presence of rickettsial DNA in ticks collected from equids and the surrounding environment in the municipalities of Uruaçu and Porangatu, located in the northern region of the state. A total of 1156 equid serum samples provided by the Goiás Agricultural Defense Agency (Agrodefesa) were analyzed for antibodies against 4 *Rickettsia* antigens. Additionally, 39 blood samples were collected from equids during a 3-day field expedition in January 2024, alongside 938 ticks collected from both animals and the environment. In total, 297 adult ticks were randomly selected for molecular analysis and tested by qPCR targeting the rickettsial *glt*A gene. Positive samples were further analyzed using cPCR to target the *omp*A and *glt*A genes. Results revealed that 9.6% (111/1156) of the serum samples were seroreactive to at least one *Rickettsia* antigen. Among these, 36% had antibodies against *Rickettsia rickettsii*, 18.9% against *Rickettsia parkeri*, 69.4% against *Rickettsia amblyommatis*, and 14.4% against *Rickettsia bellii*. Furthermore, the possible antigen responsible for a homologous reaction was found in 3.6% of equids for *R. rickettsii*, and in the same proportion for *R. bellii*, while 23.4% of animals showed antibodies for *R. amblyommatis*, and none exhibited a homologous reaction to *R. parkeri*. Meanwhile, 28.2% of the serum samples collected during the field expedition had antibodies against *R. amblyommatis*, with 72.7% identifying *R. amblyommatis* as the antigen involved in the homologous reaction. In the logistic regression analysis, the variables of education level, altitude below 500 m, and presence of female mules demonstrated a positive effect on seropositivity. Ticks from animals were identified as *Amblyomma cajennense* sensu lato, *Amblyomma sculptum*, *Dermacentor nitens*, and *Rhipicephalus microplus*, while environmental ticks were identified as *A. cajennense* s.l., *A. sculptum*, and *Amblyomma nodosum*. In the qPCR assays, two specimens of *A. cajennense* s.l., one of *A. sculptum*, and one of *D. nitens* amplified a fragment of the *glt*A gene. Of these, one *A. cajennense* s.l. specimen amplified a fragment of both the *omp*A and *glt*A genes, and one *A. sculptum* specimen amplified a fragment of the *glt*A gene through conventional PCR. Sequencing confirmed the detection of *R. amblyommatis*. These findings highlight the presence of anti-*Rickettsia* spp. antibodies in equid serum samples from all regions of the state of Goiás, emphasizing the important role of equids as sentinels for *Rickettsia* spp. To our knowledge, this study represents the largest effort to detect anti-*Rickettsia* spp. antibodies in equid serum samples in Brazil. Additionally, it is the first nationwide investigation of its kind conducted in collaboration with the Agricultural Defense Agency (Agrodefesa), serving as a significant example of the One Health approach.

## 1. Introduction

The *Rickettsia* genus is currently divided into five major groups, one of which is the spotted fever group (SFG), consisting of at least 29 species. Among these, 17 are well recognized as pathogenic to humans, with ticks being the primary vectors associated with these *Rickettsia* species [1,2,3]. This group includes *Rickettsia rickettsii*, the most pathogenic species in the genus, and *Rickettsia amblyommatis*, which was initially reported in the United States but is now widely distributed across the Americas, including Brazil. Both species are mainly transmitted by ticks of the *Amblyomma* genus [4,5,6].

Equids serve as effective sentinels for the serological detection of rickettsial exposure in regions where Brazilian spotted fever (BSF) is endemic. In fact, they can sustain high population densities of *Amblyomma sculptum* and produce a detectable humoral adaptive immune response for up to 770 days following primary exposure [7,8,9]. Sentinel animals are employed for epidemiological surveillance in various contexts and diseases [10].

The state of Goiás, located in the midwestern region of Brazil within the Cerrado biome, was considered a non-endemic area for BSF until 2009 [11]. However, in 2010, the first human case was recorded in the Notifiable Diseases Information System (SINAN), and from 2010 to 2024, 24 cases were confirmed [11]. In Goiás’s Cerrado biome, the peak occurrence of *A. sculptum* nymphs coincides with the period when human cases of BSF were reported, suggesting a connection between the seasonality of this tick stage and the transmission of human rickettsiosis in the state [12].

Owing to the limited research on the epidemiology of BSF in the Cerrado biome, this study aimed to investigate the presence of anti-*Rickettsia* spp. antibodies in georeferenced equid serum samples from all regions of Goiás. Additionally, it sought to identify variables linked to risk factors for *Rickettsia* circulation and assess the presence of rickettsial DNA in ticks collected from equids and the surrounding environment in the municipalities of Uruaçu and Porangatu, located in the northern part of the state. Notably, this is the first state-level investigation of spotted fever in Brazil using equids as sentinels and the first national level study conducted in partnership with the Agricultural Defense Agency (Agrodefesa) focused on BSF. This research exemplifies the application of a One Health approach to understanding the epidemiology of BSF in endemic and at risk areas.

## 2. Materials and Methods

### 2.1. Study Locations and Sampling Calculation

The state of Goiás spans a geographic area of 340,242.859 km^2^ and has an equine population of 390,924 animals [13], making it the sixth largest in the country. Minas Gerais is the leading producer with 788,064 equines, while the municipality of Nova Crixás is notable as the top producer within Goiás [13]. Goiás has a tropical savanna climate, characterized by dry winters and rainy summers, with warmer temperatures during the rainy season, particularly in the western and northern regions of the state. The state is predominantly covered by the Cerrado biome, the second largest biome in South America and recognized as the most biodiverse tropical savanna in the world. The Cerrado encompasses about 24% of Brazil’s territory, extending over approximately 2036.448 km^2^. Despite its immense ecological importance, only 6.5% of its original vegetation is protected [13,14,15,16].

Serum samples from equids (horses, mules, and donkeys) were collected between November 2020 and January 2021 by the Goiás Agricultural Defense Agency (Agrodefesa) to determine the prevalence of Equine Infectious Anemia Virus (EIAV) in equid herds across Goiás [17]. The animals were categorized based on their management systems: Stratum 1 included farms with only equids, Stratum 2 encompassed farms with both equids and cattle, and Stratum 3 consisted of urban equine farms with equids [17]. The animals on these farms served various purposes, such as leisure, sport, and draft power (e.g., cart horses). Farms were randomly selected from the list of equid farms registered with Agrodefesa, and the sample size was determined to ensure at least 90% specificity at the herd level [17]. A total of 330 farms were selected. On farms with up to nine equids, blood serum samples were collected from all animals, while on farms with ten or more equids, samples were taken from nine animals, prioritizing those that had been on the farm the longest [17].

Following the serology results of samples provided by Agrodefesa, a three-day field expedition was carried out in January 2024 in the northern region of Goiás, specifically in the municipalities of Uruaçu and Porangatu (Table 1). These locations were selected based on the number of sampled equids and the detection of antibodies for *Rickettsia* spp. through the indirect immunofluorescence assay (IFA). Farms that had previously participated in the EIAV serological survey were chosen for this collection, with sampling focused on the animals that had been on the farm the longest, ensuring that some had been part of the earlier study. Blood samples and ticks were collected from these animals when present. Additionally, the flannel cloth-dragging technique was used for convenience on the visited farms or in areas where the presence of capybaras had been reported.

This study was previously approved by the Institutional Animal Care and Use Committee (CEUA/UFG) of the Federal University of Goiás (Protocol No. 072/21).

### 2.2. Serological Testing

Equid serum samples were individually tested using an indirect immunofluorescence assay (IFA) targeting four *Rickettsia* antigens isolated from Brazil: *R. rickettsii* strain Pampulha, *Rickettsia parkeri* strain Atlantic rainforest, *Rickettsia amblyommatis* strain Ac37, and *Rickettsia bellii* strain Mogi, as previously described [18,19,20] and produced by the Parasitic Diseases Laboratory (LADOPAR) of the School of Veterinary and Animal Science (EVZ) at the Federal University of Goiás (UFG). For this purpose, individual sera were diluted in two-fold increments with phosphate-buffered saline (PBS) starting from an initial dilution of 1:64. Slides were incubated with fluorescein-isothiocyanate-labeled rabbit anti-horse IgG at a 1:400 dilution (Sigma, St. Louis, MO, USA). For each sample, the endpoint IgG titer reacting with the four *Rickettsia* antigens was determined. If a serum reacted to one *Rickettsia* species with an endpoint titer at least four-fold higher than the endpoint titers observed for the other *Rickettsia* species, the highest titer was considered probably homologous to the first *Rickettsia* species or a very similar species [19,21]. On each slide, a non-reactive (negative control) serum and a known reactive (positive control) serum were tested at the 1:64 dilution.

### 2.3. Statistical and Risk Factors Analysis

Statistical analysis was conducted on the results of samples provided by Agrodefesa. During farm visits, an epidemiological questionnaire was applied to farmers to gather information on various aspects of the equid populations, including the number of animals, age, breed, sex, length of stay on the farm, geographical coordinates, and environmental factors, such as water sources and forest fragments, according to previously published data [17]. This approach enabled the identification of a potential correlation between the serological results and the data gathered from the questionnaire. By presenting the variables that indicate risk factors for the circulation of *Rickettsia* in equids within the state of Goiás and beyond, it also highlights the potential risk of human infection.

A random forest model was trained on the dataset to assess the mean decrease Gini (MDG) for each predictor, taking into account non-linear relationships and interactions between variables. The predictors were ranked based on their MDG values, and the top-ranked variables were selected as predictors in the logistic regression model. This approach aimed to improve the robustness and accuracy of the predictive model.

Before conducting the multivariate analysis, a univariate analysis was performed to understand how each variable contributes to the dependent variable (seropositivity). The multivariate analysis relied on logistic regressions within a model that included variables with MDG > 2. The data were randomly split into a training set (80% for building a predictive model) and a test set (20% for evaluating the model). A seed was set for reproducibility. The Wald statistic was utilized to test the significance of individual coefficients in the model. All analyses were carried out in the R language and environment for statistical computing [22].

### 2.4. Blood Samples and Ticks

During the field collection, blood samples were obtained from equids through jugular venipuncture. Immediately after collection, the blood was placed in tubes without an anticoagulant and refrigerated until visible clot retraction occurred. The samples were then centrifuged at 1500× *g* for 10 min. The resulting serum was transferred to 2.0 mL tubes and stored at –20 °C until further processing.

Each equid underwent a thorough inspection of its entire body for ticks by two observers for five minutes. Host-seeking ticks in the environment were collected using the cloth-dragging technique, as well as by visually searching for ticks on vegetation, as previously described [23,24]. The dragging procedure involved four individuals passing cotton flannel over the undergrowth while also visually scanning the surrounding vegetation for 30 min. The locations from which ticks were collected, both from the animals and the environment, are shown in Figure 1.

Ticks were removed using anatomical tweezers and stored in 50 mL conical tubes filled with isopropyl alcohol, kept at room temperature until taxonomic identification at the LADOPAR/EVZ/UFG. The tubes were identified with the date, location, and the names of the animal and farm. Tick identification was performed to the species level using a stereomicroscope, based on descriptions and taxonomic keys [25,26,27,28,29]. Due to the lack of specific taxonomic references for *Amblyomma* larvae in Brazil, their identification was limited to the genus level [30].

As described in a previous study on the geographical distribution of the *Amblyomma cajennense* s.l. complex [28], the municipality of Porangatu, one of the locations where field expeditions for this study were conducted, represents an area of sympatry between *Amblyomma sculptum* and *Amblyomma cajennense* s.s. Since these male species can only be differentiated through molecular analysis, the male ticks were identified as *Amblyomma cajennense* s.l. for this study.

### 2.5. DNA Extraction and Molecular Detection of Rickettsia

The ticks collected in this study were randomly selected and processed individually for DNA extraction using the guanidine isothiocyanate and phenol/chloroform technique [31].

Ticks’ DNA samples were tested using a TaqMan real-time qPCR assay that targets a 147 bp fragment of the rickettsial *glt*A gene [32,33]. Positive qPCR samples were further analyzed with a conventional PCR panel targeting rickettsial genes, utilizing primers for a 532 bp fragment of the *omp*A gene and a 401 bp fragment of the *glt*A gene [32,34,35]. For each PCR run, a negative control (water) and a positive control (*Rickettsia rhipicephali*) were included.

Conventional PCR products, specifically the *omp*A and *glt*A genes, were purified using the Wizard^®^ SV Gel and PCR Clean-Up System (Promega, Madison, WI, USA) following the manufacturer’s instructions, and then sequenced using the BigDye™ Terminator v3.1 Matrix Standards Kit (Applied Biosystems, Foster City, CA, USA) at the Núcleo de Plataformas Tecnológicas (Fiocruz PE). Sequencing reactions were conducted in both directions on a 3500xL Genetic Analyzer (Applied Biosystems, Foster City, CA, USA) employing the same primers utilized for the conventional PCR assays. The resulting sequences were analyzed using Basic Local Alignment Search Tool (BLASTn; http://blast.ncbi.nlm.nih.gov/Blast.cgi, accessed on 1 April 2025) to identify the closest matches to available *Rickettsia* spp. sequences in GenBank.

Samples that tested negative for qPCR were subsequently analyzed using PCR protocols targeting the *16S rDNA* gene of ticks [36] to validate the presence of viable DNA in the extraction protocol. If a sample did not yield any product during these PCR assays, it was discarded.

## 3. Results

### 3.1. Collected Samples and Serology of Equids—Samples from Agrodefesa

Serum samples were collected from 1156 equids across 143 municipalities in the state of Goiás. Of the 330 farms sampled, 41.8% (138/330) belong to Stratum 1, 42.1% (139/330) to Stratum 2, and 16.1% (53/330) to Stratum 3. All samples were tested by IFA against four rickettsial antigens.

A total of 9.6% (111/1156) equids’ (1015 horses, 132 mules, and 9 donkeys) serum samples reacted to one or more rickettsial antigens. Among the positive serum samples, 36% (40/111) reacted to *R. rickettsii* (endpoint titer: 64–512), 18.9% (21/111) to *R. parkeri* (endpoint titer: 64–512), 69.4% (77/111) to *R. amblyommatis* (endpoint titer: 64–512), and 14.4% (16/111) to *R. bellii* (endpoint titer: 64–256), Figure 2. Furthermore, the possible antigen involved in a homologous reaction (PAIHR) was identified in four equids, representing 3.6% (4/111) for *R. rickettsii* and the same number of animals for *R. bellii*, while 23.4% (26/111) of animals reacted to *R. amblyommatis*. None of the animals showed a homologous reaction to *R. parkeri*.

Among horses, 9% (91/1015) reacted to at least one *Rickettsia* antigen, with 42.9% (39/91) for *R. rickettsii*, 23.1% (21/91) for *R. parkeri*, 63.7% (58/91) for *R. amblyommatis*, and 15.4% (14/91) for *R. bellii*. For mules, 15.15% (20/132) reacted to at least one *Rickettsia* antigen, with 5% (1/20) for *R. rickettsii*, 95% (19/20) for *R. amblyommatis*, and 10% (2/20) for *R. bellii*. No mules reacted to *R. parkeri*. Donkeys did not react to any of the four tested *Rickettsia* spp. antigens (Table 2).

### 3.2. Risk Factors

The top-ranked variables according to the MDG values (used as predictors in the logistic regression model) are found in Figure 3. In the univariate analysis, it was observed that education level (OR = 4.1; 95% CI = 1.6–12.1), altitude < 500 m (OR = 9.0; 95% CI = 1.7–67.8), purpose of herd (OR = 8.2; 95% CI = 1–79.4), buying animals in other municipalities (OR = 1.6; 95% CI = 1–2.9), presence of female mules under 6 months of age (OR = 2.6; 95% IC = 1.2–5.4) and of male mules under 6 months of age (OR = 3.4; 95% IC = 1.3–8.8), presence of water collections on the property (OR = 1.7; 95% IC = 1–2.4), equid breeding (OR = 1.6; 95% IC = 1–3.2), and presence of vegetation (OR = 1.3; 95% IC = 1–1.8) were factors that positively affected the dependent variable.

In the logistic regression (R-squared: 0.86; AUC: 0.75), the best model included the variables education level (OR = 2.9; 95% CI = 1–9.2), altitude < 500 m (OR = 14.9; 95% CI = 2.5–86.9), and presence of female mules (OR = 2.2; 95% CI = 1.1–5.1 (*p* < 0.05)). The R-squared of 0.86 indicates that the logistic regression model explained a substantial amount of the variation in the outcome variable, suggesting a strong fit. The ROC AUC of 0.75 indicates that the logistic regression model had good discriminatory ability. It suggests that the model was able to distinguish between the classes better than random chance.

### 3.3. Tick Identification and Serology of Equids—Field Expedition

Due to the higher number of equids sampled and the increased detection of antibodies to *Rickettsia* spp. through IFA, the municipalities of Uruaçu and Porangatu were selected for the field expedition. Farms were chosen based on their prior participation in the EIAV serological survey, with sampling focused on animals that had been on the farm the longest, ensuring continuity with the previous study. During this expedition, a total of 43 equids were examined, and 65.12% (28/43) were found to have tick infestations. In four locations, both the cloth-dragging technique and visual search methods were employed to capture ticks from the environment, and three of these sites also reported the presence of capybaras.

A total of 938 ticks were collected, with 73.0% (683/938) from equids and 27.0% (255/938) from the environment. Among the total ticks collected, 9.9% (93/938) were larvae, 10.5% (99/938) were nymphs, and 79.5% (746/938) were adults (Table 3).

In a fortuitous collection carried out on a puppy from one of the farms, we managed to remove 74 fleas (63 females and 11 males), identified as *Ctenocephalides felis.*

As highlighted in a previous study describing the geographical distribution of the *Amblyomma cajennense* s.l. complex [28], the municipality of Porangatu—one of the field expedition sites for this study—represents a zone of sympatry between *Amblyomma sculptum* and *Amblyomma cajennense* s.s. As male specimens cannot be reliably differentiated without molecular analysis, they were classified as *Amblyomma cajennense* s.l. The male ticks collected in this municipality and deposited in the “Coleção Nacional de Carrapatos do Cerrado Marcelo Bahia Labruna”, identified as belonging to the *Amblyomma* genus, were likewise recorded as *Amblyomma cajennense* s.l.

The following tick specimens were deposited in the “Coleção Nacional de Carrapatos do Cerrado Marcelo Bahia Labruna” (CNCC) at the School of Veterinary and Animal Science at the Federal University of Goiás: (CNCC 085) 85 larvae, 38 nymphs, and 71 adults of *D. nitens*; (CNCC 086) 3 nymphs and 1 adult of *D. nitens*; (CNCC 087) 1 larvae, 2 nymphs, and 8 adults of *D. nitens*; (CNCC 088) 6 larvae, 54 nymphs, and 118 adults of *D. nitens*; (CNCC 089) 1 adult of *R. microplus*; (CNCC 090) 1 larva of *Rhipicephalus* sp. and 5 adults of *R. microplus*; (CNCC 091) 40 adults of *A. cajennense* s.l.; (CNCC 092) 9 adults of *A. cajennense* s.l.; (CNCC 093) 20 adults of *A. sculptum*; (CNCC 094) 32 adults of *A. sculptum*; (CNCC 095) 11 adults of *A. sculptum*; (CNCC 096) 5 adults of *D. nitens,* 3 adults of *A. cajennense* s.l., and 1 adult of *A. sculptum;* (CNCC 097) 1 nymph and 14 adults of *A. sculptum;* (CNCC 098) 29 adults of *A. cajennense* s.l., 1 nymph, and 25 adults of *A. sculptum*; (CNCC 099) 12 adults of *A. cajennense* s.l., 12 adults of *A. sculptum,* and 1 adult of *A. nodosum;* (CNCC 100) 10 adults of *A. cajennense* s.l. and 21 adults of *A. sculptum*.

A total of 28.2% (11/39) of equine serum samples tested positive for *Rickettsia amblyommatis*, and of those, 72.7% (8/11) identified *Rickettsia amblyommatis* (endpoint titer: 128–512) as the antigen involved in the homologous reaction (PAIHR). None of the animals showed reactivity to *R. rickettsii*, *R. parkeri*, or *R. bellii*.

### 3.4. Molecular Detection of Rickettsia spp.

A total of 297 adult ticks, including 140 *A. cajennense* s.l., 100 *A. sculptum*, and 57 *D. nitens*, were randomly selected for molecular analysis. Of these, 1.3% (4/297) tested positive for the *glt*A gene by qPCR, comprising two *A. cajennense* s.l.—one collected from a flannel drag and one from an animal—one *A. sculptum* (collected from a flannel drag), and one *D. nitens* (from an animal), all from the municipality of Porangatu in the northern region of Goiás. Following this, conventional PCR assays targeting rickettsial genes revealed amplification of both *omp*A and *glt*A gene products from one *A. cajennense* s.l. specimen (originating from an animal), and of the *glt*A gene product from one *A. sculptum* specimen (originating from the environment).

Through sequencing, the *A. cajennense* s.l. specimen (originating from an animal) was identified as *A. sculptum*, with the *16S rRNA* sequence showing a high percentage identity (99.5% with 99% coverage) to *Amblyomma sculptum* (GenBank accession number: MT974144.1). The *omp*A sequences obtained from this tick showed a high percentage identity (99.7% with 100% coverage) to the *Rickettsia amblyommatis* strain Ac37 (GenBank accession number: CP012420.1). Attempts to sequence the *glt*A gene product of the *A. sculptum* collected from the environment were unsuccessful. GenBank accession numbers for the DNA partial sequences generated in the present study are PV394075 for the *16 S rRNA* gene of *A. sculptum* and PV407319 for the *omp*A gene of *R. amblyommatis*.

## 4. Discussion

The present study provided serological evidence that equids from the Cerrado biome have been exposed to rickettsiae. Due to the possibility of cross-reactivity among different *Rickettsia* species, it is still not possible to definitively determine which agent is associated with BSF cases in Goiás [37]. However, considering that the reported human cases were not fatal [11], the presence of a *Rickettsia* species other than *R. rickettsii* was suggested. Additionally, the study identified six tick species through taxonomic keys and confirmed the presence of *R. amblyommatis*, belonging to the SFG, in *A. sculptum* ticks in the northern region of Goiás.

Seropositivity for *Rickettsia* spp. was detected in all regions of the state. Among the equid serum samples provided by Agrodefesa, 9.60% were found to be seropositive, whereas in field-collected samples, this rate was 28.2%, indicating that these animals had been exposed to rickettsial agents at some point in their lives. These findings are consistent with previous studies that reported seroreactivity rates of 58.9%, 39.9%, and 24.6% in non-endemic areas of the Baixada Maranhense in Maranhão state [38], São Paulo state [39], and the rural zone of Ilhéus, Bahia state [40]. Furthermore, the results of this study align with research conducted in Goiás involving other animal species, such as dogs [41,42], wild boars (*Sus scrofa*) [43], small mammals, and humans [42], which also showed seropositive samples.

Only 3.60% of the positive samples for *R. rickettsii* (23.4%) in this study exhibited titers four times higher than those of other tested *Rickettsia* species, presenting low titers (128–256). This finding is consistent with studies conducted in non-endemic regions of the state of São Paulo, in which no homologous serological titers against *R. rickettsii* antigens were detected in equines [31,44].

In the present study, *R. amblyommatis* was the most prevalent species. Among the samples provided by Agrodefesa, the positivity rate for *R. amblyommatis* was 69.4%, with 23.4% of these samples exhibiting homology, while in field-collected samples, this rate was 72.7%. These results align with findings from other studies conducted in the northeastern region of Brazil, which reported higher seropositivity for *R*. *amblyommatis* in serum samples from equines and dogs [38,45]. Additionally, in the state of Mato Grosso, in midwestern Brazil, *R. amblyommatis* exhibited homology in 26.5% of equine serum samples [46]. Previous studies conducted in Panama also detected antibodies against *R. amblyommatis* in serum samples from dogs and horses, as well as the identification of this agent in ticks [47].

Despite the widespread distribution of *R. amblyommatis*, its clinical and epidemiological significance remains unclear; however, substantial scientific evidence suggests that *R. amblyommatis* causes milder pathogenicity in humans and animals [6]. In this study, *R. amblyommatis* was detected in *A. sculptum*, corroborating data indicating its presence in 34 tick species, most of which belong to the *Amblyomma* genus, across 17 countries in the Americas [6]. In Panama, this *Rickettsia* species is the most common, having been molecularly detected in nine tick species and isolated from *A. mixtum* for the first time in 2021 [48]. It is suggested that *R. amblyommatis* is the most prevalent and widely distributed SFG rickettsia in the Americas [49]. Studies propose that *R. amblyommatis* infection may reduce the severity of BSF when co-infection with *R. rickettsii* occurs. This interaction may relate to competition between microorganisms within ticks or the immune response generated in the infected host [6].

Due to the ease of handling equines for blood collection [44] and their persistent immune response, where anti-*R. rickettsii* antibodies have been detected for up to two years post-infection [9], these animals can serve as effective sentinels for detecting the circulation of *R. rickettsii* in areas where *A. sculptum* uses equines as hosts [9]. The results of this study indicate that investigating rickettsial agents in equid serum samples is essential for understanding the true epidemiological situation of spotted fever in the Cerrado biome. This finding aligns with previous studies suggesting that in locations where humans are exposed to *A. sculptum* ticks, the use of equines as sentinels could help identify risk areas before human cases arise [31]. Serological testing in equines plays a crucial role in the active surveillance of BSF, as data on the seroprevalence of these animals can enhance human diagnosis and monitoring in areas without reported human cases but with ecological conditions conducive to *Rickettsia* spp. transmission [39].

Regarding statistical and risk factor analysis, MDG was utilized for variable selection by pre-screening variables before fitting the logistic regression model, which is especially beneficial when dealing with a large number of potential predictors. This approach aimed to reduce overfitting by excluding irrelevant or weakly predictive variables, thereby enhancing the logistic regression model’s ability to generalize and potentially improving performance metrics, such as accuracy [50]. Additionally, logistic regression can be sensitive to multicollinearity (high correlation between predictors), which can inflate standard errors and complicate interpretation; thus, MDG offers a means to prioritize variables that are individually significant predictors, potentially alleviating issues caused by multicollinearity [51].

On this subject, the present study demonstrated that the most significant variables were the education level of the animal caretakers, altitudes below 500 m, and the presence of female mules. To a lesser but still considerable extent, the variables of herd purpose, animals purchased from other municipalities, presence of male mules, presence of water sources, and vegetation showed a positive correlation with the presence of *Rickettsia* spp. Previous research has shown that the tick *A. sculptum* is more prevalent in low-altitude and high-humidity areas, where it finds optimal conditions for its lifecycle and parasitism [24]. Another study observed a positive correlation with the animals’ age and the duration of their stay on the property. This is supported by findings from earlier research, which indicated that animals over 12 years old were 2.1 times more likely to be seropositive, suggesting that older animals were more exposed throughout their lives [52]. Additionally, due to the prolonged persistence of antibodies, selecting horses to serve as sentinels should consider those born or residing in the area for over two years [9].

It is important to highlight the relevance of this study in several ways. Firstly, we used a large sample size, which included the analysis of 1156 equid serum samples tested for 4 *Rickettsia* spp. antigens. Furthermore, the study covered all regions of Goiás, where the Cerrado biome is predominant, and it also marks the first statewide investigation to employ equids as sentinels for SFG rickettsiae. Our study highlights the practicality of utilizing samples from animal health surveillance agencies for research and efforts in zoonosis prevention, thereby fostering collaboration consistent with the principles of the One Health approach. This study model may be replicated in other areas.

## 5. Conclusions

In conclusion, this study revealed the circulation of rickettsial agents in serum samples and ticks from equids in the Cerrado biome of Goiás, as evidenced by the detection of *Rickettsia amblyommatis* in *Amblyomma sculptum* in the northern region of the state. The observed prevalence reinforced the role of equids as effective sentinels and highlighted their epidemiological importance in understanding spotted fever in the Cerrado biome. Additionally, variables associated with risk factors for the circulation of *Rickettsia* in equids were identified, including education level, location at altitudes below 500 m, and the presence of female mules, which may pose a potential risk for human infection. To our knowledge, this is the first national study conducted in partnership with the Goiás Agricultural Defense Agency and represents the largest global effort focused on detecting anti-*Rickettsia* spp. antibodies in equid samples. This work stands out as a model of the One Health approach.

## Figures and Tables

**Figure 1 pathogens-14-00449-f001:**
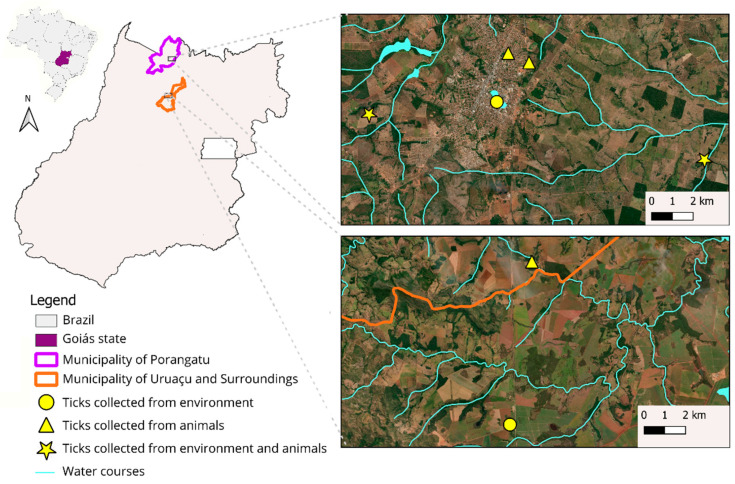
Geographic locations where tick collections were sampled in the state of Goiás, midwestern Brazil. These locations were selected based on the number of equids sampled and the detection of antibodies for *Rickettsia* spp. through IFA.

**Figure 2 pathogens-14-00449-f002:**
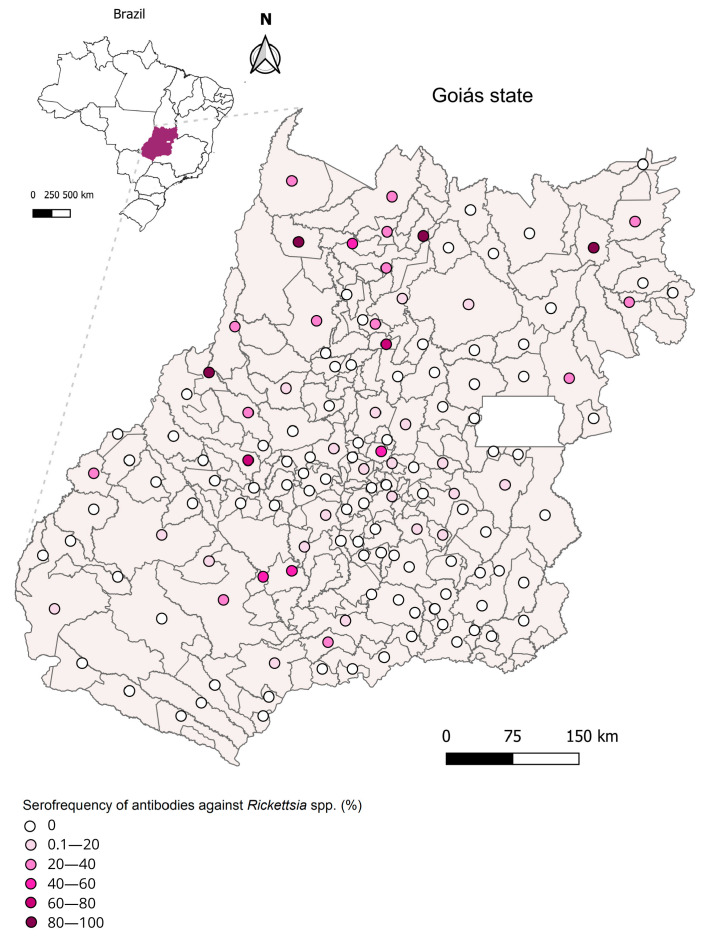
Geographical location of the surveyed municipality and serofrequency against anti-*Rickettsia* antibodies of equids from Goiás state, midwestern Brazil. All regions of the state of Goiás presented seropositivity.

**Figure 3 pathogens-14-00449-f003:**
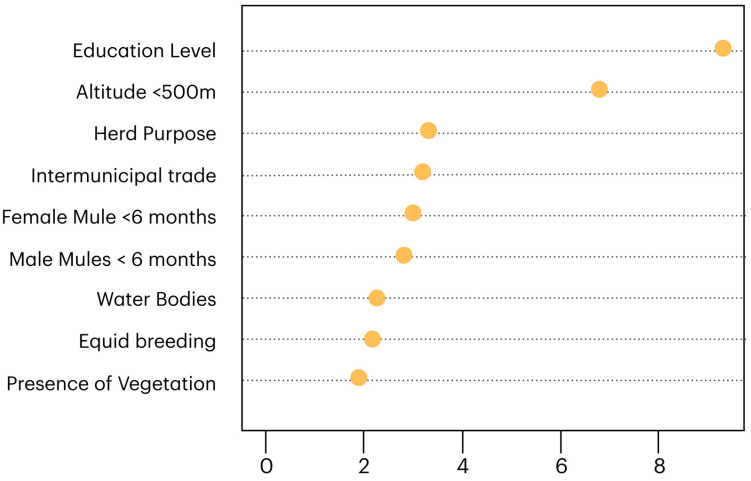
Top-ranked variables according to mean decrease Gini.

**Table 1 pathogens-14-00449-t001:** Geographical coordinates and description of the field expedition carried out in January 2024 in the municipalities of Uruaçu and Porangatu in the state of Goiás, Midwestern Brazil.

Municipality	Location	Collection in Equids	Collection in Environment	Presence of Capybaras
Uruaçu	14°21′54.50″ S and 49°09′05.80″ W	X		
Uruaçu	14°26′05.80″ S and 49°09′37.00″ W		X	X
Porangatu	13°25′54.80″ S and 49°07′46.20″ W	X		
Porangatu	13°25′40.79″ S and 49°08′14.60″ W	X		
Porangatu	13°26′35.96″ S and 49°08′32.23″ W		X	X
Porangatu	13°27′03.59″ S and 49°11′22.74″ W	X	X	X
Porangatu	13°28′00.66″ S and 49°03′54.76″ W	X	X	

**Table 2 pathogens-14-00449-t002:** Seroreactivity to four *Rickettsia* species of equids from Goiás state, midwestern Brazil, a non-endemic area for Brazilian spotted fever, from November 2020 to January 2021.

Municipalities with Positive Samples	No. of Seroreactive Animals to Each *Rickettsia* Species (No. of Positives/No. of Animals Tested; % Seroreactivity)				No. Animals with PAIHR *^a^*
	*R. rickettsii*	*R. parkeri*	*R. amblyommatis*	*R. bellii*	
Abadiânia	0 (0)	0 (0)	0 (0)	1 (1/7; 14.3)	1 R.be *^b^*
Acreúna	2 (2/6; 33.3)	4 (4/6; 66.7)	2 (2/6; 33.3)	0 (0)	
Amaralina	1 (1/12; 8.3)	1 (1/12; 8.3)	5 (5/12; 41.7)	0 (0)	2 R.am ^c^
Aruanã	0 (0)	0 (0)	1 (1/3; 33.3)	0 (0)	1 R.am
Baliza	0 (0)	1 (1/8; 12.5)	2 (2/8; 25)	0 (0)	
Bela Vista de Goiás	2 (2/6; 33.3)	0 (0)	0 (0)	0 (0)	
Bom Jesus de Goiás	1 (1/3; 33.3)	1 (1/3; 33.3)	0 (0)	1 (1/3; 33.3)	
Britânia	1 (1/1; 100)	1 (1/1; 100)	1 (1/1; 100)	1 (1/1; 100)	
Caiapônia	1 (1/2; 50)	0 (0)	0 (0)	0 (0)	
Campo Limpo de Goiás	0 (0)	0 (0)	1 (1/2; 50)	0 (0)	
Crixás	3 (3/15; 20)	0 (0)	5 (5/15; 33.3)	0 (0)	1 R.ri ^d^; 3 R.am
Faina	0 (0)	0 (0)	1 (1/9; 11.1)	0 (0)	
Fazenda Nova	0 (0)	0 (0)	2 (2/3; 66.7)	1 (1/3; 33.3)	
Formosa	7 (7/11; 63.6)	6 (6/11; 54.5)	3 (3/11; 27.3)	1 (1/11; 9.1)	1 R.ri
Formoso	2 (2/2; 100)	1 (1/2; 50)	2 (2/2; 100)	0 (0)	1 R.am
Goiânia	1 (1/4; 25)	0 (0)	0 (0)	1 (1/4; 25)	
Goiatuba	1 (1/9; 11.1)	1 (1/9; 11.1)	1 (1/9; 11.1)	1 (1/9; 11.1)	
Hidrolina	0 (0)	0 (0)	1 (1/4; 25)	0 (0)	
Inhumas	2 (2/9; 22.2)	2 (2/9; 22.2)	0 (0)	0 (0)	
Itaberaí	0 (0)	0 (0)	0 (0)	1 (1/9; 11.1)	
Itapirapuã	1 (1/3; 33.3)	1 (1/3; 33.3)	1 (1/3; 33.3)	0 (0)	
Jandaia	0 (0)	0 (0)	1 (1/9; 11.1)	0 (0)	
Jaraguá	0 (0)	0 (0)	0 (0)	1 (1/2; 50)	1 R.be
Luziânia	0 (0)	0 (0)	0 (0)	1 (1/4; 25)	
Mara Rosa	0 (0)	0 (0)	1 (1/2; 50)	0 (0)	
Mineiros	1 (1/6; 16.6)	0 (0)	1 (1/6; 16.6)	0 (0)	
Montividiu do Norte	0 (0)	0 (0)	1 (1/9; 11.1)	0 (0)	
Mundo Novo	0 (0)	0 (0)	2 (2/2; 100)	0 (0)	2 R.am
Mutunópolis	3 (3/6; 50)	2 (2/6; 33.3)	2 (2/6; 33.3)	0 (0)	
Niquelândia	0 (0)	0 (0)	1 (1/5; 20)	0 (0)	
Nova Roma	0 (0)	0 (0)	1 (1/1; 100)	0 (0)	1 R.am
Palmeiras de Goiás	1 (1/7; 14.3)	0 (0)	2 (2/7; 28.6)	0 (0)	
Pirenópolis	0 (0)	0 (0)	1 (1/7; 14.3)	0 (0)	
Porangatu	3 (3/37; 8.1)	0 (0)	19 (19/37; 51.3)	0 (0)	9 R.am
Quirinópolis	1 (1/2; 50)	0 (0)	0 (0)	0 (0)	
Rio Verde	3 (3/14; 21.4)	0 (0)	0 (0)	2 (2/14; 14.3)	1 R.ri
Santo Antônio da Barra	0 (0)	0 (0)	0 (0)	1 (1/2; 50)	1 R.be
São Domingos	0 (0)	0 (0)	2 (2/5; 40)	1 (1/5; 20)	1 R.am
São Luíz do Norte	1 (1/7; 14.3)	0 (0)	5 (5/7; 71.4)	0 (0)	2 R.am
São Miguel do Araguaia	0 (0)	0 (0)	6 (6/19; 31.6)	0 (0)	3 R.am
São Mg do Passa Quatro	0 (0)	0 (0)	1 (1/8; 12.5)	0 (0)	
Silvânia	0 (0)	0 (0)	1 (1/9; 11.1)	0 (0)	1 R.am
Simolândia	1 (1/6; 16.7)	0 (0)	1 (1/6; 16.7)	1 (1/6; 16.7)	
Taquaral de Goiás	0 (0)	0 (0)	0 (0)	1 (1/5; 20)	1 R.be
Turvelândia	0 (0)	0 (0)	1 (1/6; 16.7)	0 (0)	
Uruaçu	1 (1/9; 11.1)	0 (0)	0 (0)	0 (0)	1 R.ri
TOTAL (317)	40 (40/317; 12.6)	21 (21/317; 6.6)	77 (77/317; 24.3)	16 (16/317; 5.04)	

*^a^* PAIHR: probable antigen involved in a homologous reaction. A homologous reaction was determined when the endpoint titer to a *Rickettsia* species was at least four-fold higher than those observed for the other *Rickettsia* species. In this case, the *Rickettsia* species (or a very closely related species) involved in the highest endpoint titer was considered the PAIHR. *^b^* R.be, *R. bellii*; ^c^ R.am, *R. amblyommatis*; ^d^ R.ri, *R. rickettsii*.

**Table 3 pathogens-14-00449-t003:** Ticks collected from equids and the environment in January 2024 in the municipalities of Uruaçu and Porangatu in the state of Goiás, midwestern Brazil.

Tick Host (No. of Individuals)	Location	*D. nitens*			*A. sculptum*		*A. cajennense* s.l	*A. nodosum*	*R.* sp.	*R. microplus*
		L	N	A	N	A	A	A	L	A
Equine (10)	Uruaçu	85	38	92		37				
Environment	Uruaçu				1	19				
Equine (17)	Porangatu	7	59	161		67	118		1	6
Mule (1)	Porangatu			7		2	3			
Environment	Porangatu				1	111	122	1		
TOTAL		449			238		243	1	1	6

*D. nitens*: *Dermacentor nitens*; *A. sculptum*: *Amblyomma sculptum*; *A. cajennense* s.l.: *Amblyomma cajennense* s.l.; *A. nodosum*: *Amblyomma nodosum*; *R.* sp.: *Rhipicephalus* sp.; *R. microplus*: *Rhipicephalus microplus*. L: larvae; N: nymphs; A: adults; No.: number.

## Data Availability

The data presented in this study are available within this article.

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
