# Peer review of "Spatial Distribution of Equid Exposure to Rickettsia spp. in Goiás State, Midwestern Brazil"

_pathogens, 2025, doi:10.3390/pathogens14050449_

Round 1

Reviewer 1 Report

Comments and Suggestions for Authors

The authors of the study aimed to investigate the presence of anti-Rickettsia antibodies in georeferenced equid serum samples from all regions of Goiás. Additionally, the study sought to identify variables associated with risk factors for Rickettsia circulation and assess the presence of rickettsial DNA in ticks collected from equids and the surrounding environment in the municipalities of Uruaçu and Porangatu. This research uses equids as sentinels in a One Health approach to better understand the epidemiology of spotted fever in endemic and at-risk areas.
While the study is well-developed, there are several areas for improvement:
1.    The abstract is too lengthy and should be shortened to comply with the journal's guidelines.
2.    The introduction needs to be updated to reflect the current number of Rickettsia species identified, as this number has exceeded 25. Additionally, more recent references should be included. Furthermore, the introduction could be strengthened by incorporating the role of capybaras in the tick and pathogen cycles under investigation.
3.    Clarification on the immunofluorescence assay: Is the immunofluorescence assay homemade? Are different strains of Rickettsia species used in the assay? Also, are the slides produced in-house, or are they sourced from third parties? Please provide further details on these aspects. 
4.    In paragraph 2.5, it is important to specify the primers and probes used in the study.

Minor Issues:
5.    In lines 112-113, after the colon, please indicate Stratum 1, Stratum 2, and Stratum 3, as reported in the results section.
6.    In line 349, remove the term "spp."

Author Response

Dear Reviewer 1,

Best regards, 

Reviewer 2 Report

Comments and Suggestions for Authors

The study entitled “Spatial Distribution of Equid Exposure to Rickettsia spp. in Goiás State, Midwestern Brazil” is well organized and reported. This study reveals the circulation of rickettsial agents between equines and ticks in the Cerrado biome of Goiás, reinforcing the role of equines as effective sentinels and highlighting their epidemiological importance in understanding Brazilian spotted fever in the Cerrado biome. Only a few changes are necessary for this manuscript as indicated below.

  1. L79: Add “human” to “the first case” for smooth reading.
  2. L134-L136 [Table 1]: Different presentations of latitude and longitude for locations are mixed: sexagesimal expression and decimal expression. Please unify the presentation.
  3. L208 (section 2.5), L327 (section 3.4): “Rickettsia” in the subheadings could be in roman, because other words in roman are changed to be in italic. At L327, “spp.” in italic.
  4. L225: “Basic Local Alignment Search Tool (BLAST)” (?)
  5. L333-L336: Ticks could not amplify the genes. PCR can it.
  6. L349: Delete either “spp.” or “species” after “Rickettsia”
  7. L473-L623: Please unify the presentation of references with reference to the Information for Authors. Delete unnecessary explanation between L621 and L623.
  8. Thank you for your nice study.

Author Response

Dear Reviewer 2, 

Best regards, 
